# Increased Hippocampal-Inferior Temporal Gyrus White Matter Connectivity following Donepezil Treatment in Patients with Early Alzheimer’s Disease: A Diffusion Tensor Probabilistic Tractography Study

**DOI:** 10.3390/jcm12030967

**Published:** 2023-01-27

**Authors:** Gwang-Won Kim, Kwangsung Park, Yun-Hyeon Kim, Gwang-Woo Jeong

**Affiliations:** 1Advanced Institute of Aging Science, Chonnam National University, Gwangju 61186, Republic of Korea; 2Department of Psychiatry, Massachusetts General Hospital, Harvard Medical School, Boston, MA 02129, USA; 3Department of Urology, Chonnam National University Hospital, Chonnam National University Medical School, Gwangju 61469, Republic of Korea; 4Department of Radiology, Chonnam National University Hospital, Chonnam National University Medical School, Gwangju 61469, Republic of Korea

**Keywords:** Alzheimer’s disease, diffusion tensor imaging scalars, donepezil treatment, hippocampus-related networks, probabilistic tractography

## Abstract

The incidence of Alzheimer’s disease (AD) has been increasing each year, and a defective hippocampus has been primarily associated with an early stage of AD. However, the effect of donepezil treatment on hippocampus-related networks is unknown. Thus, in the current study, we evaluated the hippocampal white matter (WM) connectivity in patients with early-stage AD before and after donepezil treatment using probabilistic tractography, and we further determined the WM integrity and changes in brain volume. Ten patients with early-stage AD (mean age = 72.4 ± 7.9 years; seven females and three males) and nine healthy controls (HC; mean age = 70.7 ± 3.5 years; six females and three males) underwent a magnetic resonance (MR) examination. After performing the first MR examination, the patients received donepezil treatment for 6 months. The brain volumes and diffusion tensor imaging scalars of 11 regions of interest (the superior/middle/inferior frontal gyrus, the superior/middle/inferior temporal gyrus, the amygdala, the caudate nucleus, the hippocampus, the putamen, and the thalamus) were measured using MR imaging and DTI, respectively. Seed-based structural connectivity analyses were focused on the hippocampus. The patients with early AD had a lower hippocampal volume and WM connectivity with the superior frontal gyrus and higher mean diffusivity (MD) and radial diffusivity (RD) in the amygdala than HC (*p* < 0.05, Bonferroni-corrected). However, brain areas with a higher (or lower) brain volume and WM connectivity were not observed in the HC compared with the patients with early AD. After six months of donepezil treatment, the patients with early AD showed increased hippocampal-inferior temporal gyrus (ITG) WM connectivity (*p* < 0.05, Bonferroni-corrected).

## 1. Introduction

Alzheimer’s disease (AD) is characterized by a progressive deterioration in learning and memory abilities, and it typically progresses slowly in three general stages: early (mild), middle (moderate), and late (severe) [1]. Mild cognitive impairment (MCI) can be defined as cognitive decline greater than expected for individual age and education, without interfering with the activities of daily living, which is the prodromal stage of AD [2,3,4]. MCI and mild dementia due to AD represent early-stage AD [5,6,7,8,9]. Approximately 10–15% of patients with MCI progress to AD each year, whereas only 1–2% of individuals with a normal cognitive level develop AD [10,11]. The early detection of AD and intervention are essential to predict and prevent AD.

Much data have accumulated over two decades about the effects of structural abnormalities in the brain on MCI and AD [12,13,14,15,16,17,18,19,20]. Prior studies have reported that hippocampal atrophy is specifically implicated in AD. Brain atrophy is correlated with both tau deposition and neuropsychological deficits, and it is a valid marker of AD and its progression [13]. According to a meta-analysis [12], patients with MCI show a 2.2-fold higher volume loss in the hippocampus, a 1.8-fold higher volume loss in the whole brain, and a 1.5-fold higher volume loss in the entorhinal cortex than healthy controls. Accordingly, numerous studies have focused on the atrophy of the hippocampus and hippocampal subfields in AD [21,22,23,24,25,26,27,28]. A structural magnetic resonance imaging (MRI) study [25] revealed decreased gray matter (GM) volume in the hippocampus of patients with early AD, specifically in the right subiculum and left cornu ammonis (CA3). A similar study [22] suggested that decreased volumes involving the hippocampus and hippocampal-precuneus/posterior cingulate cortical tracts are associated with the early signs of AD in patients with MCI. In addition, patients with MCI have been found to have a significantly lower functional connectivity from the left hippocampus to the right inferior temporal gyrus, right middle temporal gyrus, right parahippocampal gyrus, and part of the medial frontal cortex than normal controls [11].

Treatment with acetylcholinesterase inhibitors (AChEIs) in patients with early AD prevents the breakdown of acetylcholine (ACh) and increases cholinergic transmission, resulting in improved cognitive function [25,29]. Donepezil is the most frequently prescribed drug clinically to inhibit acetylcholinesterase activity, suggesting higher ACh activity in the brain regions related to cognitive function [30,31,32]. A functional magnetic resonance imaging (fMRI) study [33] reported increased medial temporal lobe activation and increased connectivity of cholinergic networks after 3-month donepezil treatment in patients with MCI. A similar study [34] revealed increased activity in the ventrolateral prefrontal cortex during a visual memory task after 6-month donepezil treatment of MCI.

Recent studies [35] have shown that complex networks, along with diffusion tensor imaging (DTI), are effective and promising for the early detection of changes in the structural pathology of patients with AD. Probabilistic tractography in DTI enables the detection of the WM integrity of an entire bundle, facilitating the evaluation of structural connectivity by estimating the likelihood of connection between two areas of the brain [36,37]. The most prominent structural changes in AD occur initially in hippocampus, and the atrophy of this area is a diagnostic marker for AD at the mild cognitive impairment stage [13]. Leung et al. [38] suggested that the acceleration in hippocampal atrophy rates in patients with MCI seems to be driven by the concurrent progression to a clinical diagnosis of AD. A positron emission tomography (PET) study [39] reported that reduced hippocampal connectivity occurs predominantly in the AD connectome, correlating with hippocampal tau in MCI. A structural study investigating the interaction between the hippocampus and cortical/subcortical regions using probabilistic tractography following donepezil treatment has yet to be reported. Identifying objective predictors of WM connectivity in early AD can contribute to data-driven approaches aimed at AD prevention.

Thus, the purpose of this study was to evaluate alterations in the hippocampal WM connectivity in early-stage AD before and after donepezil treatment. In addition, as a secondary analysis, we also evaluated the WM integrity and changes in brain volume.

## 2. Subjects and Method

### 2.1. Participants

Ten patients with early-stage AD (mean age = 72.4 ± 7.9 years) underwent an MR examination before (baseline) and after (6-month follow-up) donepezil treatment (Appendix A). The control group included nine healthy controls (mean age = 70.7 ± 3.5 years), who were recruited via advertisements. The patients with early AD were recruited based on the following criteria [25,37,40]: (1) having possible or probable AD according to DSM-IV and the National Institute of Neurological and Communicative Diseases and Stroke-Alzheimer Disease and Related Disorders Association (NINCDS-ADRDA) criteria; (2) having no history of AD treatment or other neurological or psychiatric illnesses; (3) having a score of 0.5 or 1 on the Clinical Dementia Rating (CDR) scale; (4) having a score of less than 26 on the Korean version of the Mini-Mental State Examination (K-MMSE); and (5) having typical symptom severity, including changes in cognition recognized by the affected individual or observers, objective impairment in one or more cognitive domains, functional independence, and the absence of dementia. After performing the first MR examination, the patients received 5 mg/day of Aricept^®^ (donepezil hydrochloride; Pfizer Inc., New York, NY, USA) for the first 28 days and 10 mg/day thereafter. Donepezil hydrochloride has been approved by the Ministry of Food and Drug Safety (MFDS) in Korea. The treatment duration for the patients was 194.0 ± 29.5 days, without any side effects, such as agitation, gastrointestinal bleeding, and stomach ulcers. Healthy controls were selected based on the following criteria: (1) no AD based on both the DSM-IV and the NINCDS-ADRDA criteria; (2) a score greater than 26 on the K-MMSE; and (3) no history of AChEI treatment or neurological or psychiatric disorders.

To assess cognitive function, the patients with and without donepezil treatment were assessed using the following cognitive or neuropsychological scales: K-MMSE to determine the severity of cognitive decline; the AD assessment scale–cognitive subscale (ADAS-Cog) to establish the severity of cognitive and non-cognitive dysfunction from mild-to-severe AD; CDR to assess the severity of cognitive impairment (CDR of 0 = clinical normality; 0.5 = very mild dementia; 1 = mild dementia; 2 = moderate dementia; 3 = severe dementia); and the geriatric depression scale (GDS) to evaluate the severity of depressed mood (GDS of 0–9 = normal; 10–19 = mild depression; 20–30 = severe depression) [41]. Interviews were administered to the patients with early AD before and after the 6-month treatment. A Mann-Whitney *U*-test was used to analyze the differences between the healthy controls and the patients with early AD and between the healthy controls and the donepezil-treated patients. A Wilcoxon’s signed-rank test was used to compare the scores on K-MMSE, ADAS-Cog, CDR, and GDS between the patients with early AD and the treated patients.

### 2.2. Image Acquisition

All MRI data were collected on a 3T clinical scanner (Magnetom Tim Trio, Siemens Medical Solutions, Erlangen, Germany) using an 8-channel head coil. Sagittal T1-weighted images were acquired using a 3-dimensional magnetization-prepared rapid-acquisition gradient-echo pulse sequence with the following parameters: repetition time (TR) = 1700 ms, echo time (TE) = 2.2 ms, field of view (FOV) = 256 × 256 mm^2^, number of excitations (NEX) = 1, matrix = 512 × 512, slice thickness = 5 mm, and slice gap = 2 mm. Axial DTI were acquired using echo-planar imaging pulse sequence with the following parameters: TR: 5200 ms, TE = 105 ms, matrix = 128 × 128, FOV = 220 × 220 mm^2^, slice thickness = 5 mm, and number of slices = 30. DTI consists of 24 directions (b factor = 1000 s/mm^2^) and 5 images without diffusion weighting (b factor = 0 s/mm^2^). Phase-encoding was conducted in the anterior-to-posterior direction using a factor of 2 in-plane acceleration.

### 2.3. Data Processing and Analysis

The T1-weighted images were analyzed with FreeSurfer v6.0 software (Massachusetts General Hospital, Harvard Medical School; http://surfer.nmr.mgh.harvard.edu, 23 January 2017). The DT images were analyzed using Functional Magnetic Resonance Imaging of the Brain (FMRIB) Software Library (FSL) v6.0 software (Oxford, UK; www.fmrib.ox.ac.uk/fsl, 20 October 2018). The Enhancing Neuroimaging Genetics through Meta-Analysis (ENIGMA) protocol was used to detect outliers and for visual inspection.

#### 2.3.1. Brain Volume Analysis

The post-processing of the T1 images entailed the following steps using the FreeSurfer segmentation pipeline [25,42]: correction for head motion and the non-uniformity of intensity; the Talairach transformation of each subject’s brain; the removal of non-brain tissue; the segmentation of cortical gray matter (GM), subcortical white matter (WM), and deep GM volumetric structures; the triangular tessellation of the GM/WM interface and the GM/cerebrospinal fluid boundary; and topology correction. Based on previous studies focused on AD, the brain regions of interest (ROIs) were selected as follows: the superior/middle/inferior frontal gyrus (SFG/MFG/IFG), the superior/middle/inferior temporal gyrus (STG/MTG/ITG), the amygdala, the caudate nucleus, the hippocampus, the putamen, and the thalamus (Figure 1). These ROIs were extracted for individual T1 imaging via the automated parcellation of FreeSurfer. A Mann-Whitney *U*-test was used to compare the brain volumes between the healthy controls and the patients with early AD, and a Wilcoxon signed-rank test was used to compare the brain volumes between patients treated with and without donepezil using SPSS (version 27.0, IBM Corp., Armonk, NY, USA). The significance level was set to 0.05 (*p* < 0.0046) after Bonferroni correction for the 11 ROIs to adjust for multiple comparisons.

#### 2.3.2. DTI Scalars and WM Connectivity Analyses

DTI pre-processing entailed skull removal (bet) and correction for motion and eddy currents [37]. Multiple DTI scalars (FA; fractional anisotropy, MD; mean diffusivity, RD; radial diffusivity, and AD; axial diffusivity) were generated for individual subjects using the DTIFIT program, which fits a DT model at each voxel of the diffusion images. The individual T1 images were rigidly registered to their corresponding non-diffusion-weighted (B0) images using FMRIB’s Linear Image Registration Tool (FLIRT) combined with the mutual information cost function and trilinear interpolation [37]. The 11 ROIs were extracted for each hemisphere in each subject’s T1 imaging data via automated parcellation.

One patient showed motion artifacts in the T1 images obtained after treatment, and, thus, 11 ROIs in the patient were extracted in the T1 image obtained before treatment to register their T1 images with the diffusion space. We calculated the average values of FA, MD, RD, and AD in the 11 ROIs of the 3 groups. To evaluate the structural connectivity, diffusion parameters were modeled using the Bayesian Estimation of Diffusion Parameters Obtained using Sampling Techniques (BEDPOSTX) with crossing-fiber modeling [37]. The BEDPOSTX model of diffusion signals as ball (isotropic) and stick (anisotropic) components generates a distribution of likely fiber orientations within each voxel, as well as an estimate of the uncertainty in these orientations [43]. We used FSL probabilistic tractography (connectivity modeling) to evaluate the WM connectivity between the seed (hippocampus) and target (10 ROIs) regions as follows: 5000 streamlines per each voxel in the thalamus, a 0.2 curvature threshold, a 0.5 mm step length, and loop check. The connectivity values were routinely thresholded at 10% in order to eliminate aberrant connections due to noise and error [37,44]. A Mann-Whitney *U*-test was used to compare the DTI scalars and WM connectivity between the healthy controls and the patients with early AD. Wilcoxon’s signed-rank test was used to compare the DTI scalars and WM connectivity between the patients treated with and without donepezil using SPSS (version 27.0, IBM Corp., Armonk, NY, USA). The significance level was set to 0.05 (*p* < 0.0046 for DTI scalars, and *p* ≤ 0.005 for WM connectivity) after Bonferroni correction for the 10 to 11 ROIs in order to adjust for multiple comparisons.

## 3. Results

### 3.1. Symptom Severity

The mean K-MMSE scores in the patients with early AD (baseline), the treated patients (follow-up), and the healthy controls were 16.5 ± 4.9, 17.5 ± 2.9, and 28.6 ± 1.1, respectively (Appendix A). The mean K-MMSE score of the patients with early AD was improved by approximately 8% after treatment (*p* < 0.05). This finding is consistent with prior findings [25,45,46,47] of improved MMSE scores in patients with MCI and early AD after donepezil treatment, suggesting that donepezil treatment is associated with improved cognitive performance. The mean ADAS-Cog scores in the patients with early AD and the treated patients were 25.6 ± 6.2 and 24.4 ± 5.9, respectively (*p* > 0.05); the mean CDR scores were 0.6 ± 0.2 and 0.6 ± 0.2, respectively (*p* < 0.05); and the mean GDS scores were 13.2 ± 5.2 and 12.7 ± 4.9, respectively (*p* < 0.05). The mean ADAS-Cog and GDS scores of the patients with early AD declined by 4.7% and 3.8% after treatment, respectively. However, these results were not significantly different (*p* > 0.05) between before and after treatment.

### 3.2. Changes in Brain Volume

The patients with early AD showed a significantly decreased hippocampal volume compared with the healthy controls (*p* < 0.05, Bonferroni-corrected) (Figure 2 and Figure 3, Table 1). However, no significant difference (*p >* 0.05) was detected in the 11 ROIs before and after treatment in the group of patients with early AD (Figure 3 and Table 1).

### 3.3. Changes in DTI Scalars

Compared with the healthy controls, the patients with early AD had a higher MD and RD in the amygdala (*p* < 0.05, Bonferroni-corrected) (Figure 4, Appendix A). The patients with early AD had a lower FA in the hippocampus and amygdala (*p* ≤ 0.05, not corrected for multiple comparisons) (Figure 2, Appendix A). None of the other ROIs showed significant differences in MD and RD between the healthy controls and the patients with early AD (Appendix A). In addition, no significant difference (*p >* 0.05) was detected in the DTI scalars of the 11 ROIs before and after treatment in the group of patients with early AD (Appendix A).

### 3.4. Hippocampal White Matter Connectivity

The patients with early AD showed a significant decrease in hippocampal-SFG WM connectivity compared with the healthy controls (*p* < 0.05, Bonferroni-corrected) (Figure 2 and Table 2). Following 6-month donepezil treatment, the patients with early AD showed increased hippocampal-ITG WM connectivity (*p* < 0.05, Bonferroni-corrected) (Figure 5 and Table 2). There was no significant correlation between the changes in MMSE scores and hippocampal-ITG WM connectivity before and after donepezil treatment (*p* > 0.05).

## 4. Discussion

### 4.1. Summary of Main Findings

The patients with early AD had a lower hippocampal volume and WM connectivity with the SFG and higher MD and RD in the amygdala than healthy controls (*p* < 0.05, Bonferroni-corrected). The hippocampal volume loss is consistent with the evidence supporting AD diagnosis and tracking [48,49]. Further, the patients with early AD showed enhanced MMSE scores and increased hippocampal-ITG connectivity (*p* < 0.05, Bonferroni-corrected) after the 6-month donepezil treatment. These results suggest that the increased hippocampal-ITG WM connectivity may be attributed to donepezil treatment.

### 4.2. Brain Volume and DTI Scalars in Early AD

It is well-known that hippocampus atrophy is at the core of AD pathophysiology. The patients with early AD showed a significant decrease in hippocampal volume compared with the healthy controls. These results support the notion that hippocampal abnormalities are associated with the early detection of AD [14,22,50,51,52]. However, no volumetric increase across all the brain areas was detected after the 6-month donepezil treatment.

The patients with early AD had a higher MD and RD in the amygdala than healthy controls (*p* < 0.05, Bonferroni-corrected). The patients with early AD had a lower FA in the hippocampus and amygdala (*p* < 0.05, not corrected for multiple comparisons), but the level of significance obtained via a multiple comparison correction was not high enough to validate this finding. A DTI study [53] reported a decreased FA and a three-fold increased trace value compared with MD in the hippocampus and amygdala of patients with AD compared with healthy controls. A similar study [15] also found a significantly elevated MD in the hippocampus and amygdala of patients with AD. MD measures the average diffusivity in the non-colinear directions of free diffusion, and RD quantifies the diffusion of water molecules in a direction perpendicular to the axon fibers [54,55,56]. The increased MD in the amygdala of patients with early AD has been found to be associated with an increase in free water diffusion, and the increased RD has been found to be related to greater myelin damage. Thus, alterations in hippocampal volume and DTI scalars in patients with early AD may be associated with the early prediction of progression to AD.

### 4.3. Structural Connectivity in Early AD

Hippocampal-cortical cortex connectivity is potentially important for the early diagnosis of AD. The hippocampus is a major part of the common declarative memory network, suggesting that this brain region plays a key role in both semantic and episodic memory [57,58]. Patients with semantic dementia have been found to show decreased hippocampal-lingual gyrus functional connectivity compared with patients with AD and healthy controls [58]. Strikingly, it has also been found that when the functional connectivity between the right hippocampus and the medial prefrontal cortex (PFC) is disrupted in patients with early AD, the connectivity between the left hippocampus and the right lateral PFC increases [59,60]. We found decreased hippocampal-SFG (medial PFC) WM connectivity in the patients with early AD compared to the healthy controls. This result, which has not been reported in any previous structural connectivity studies, is consistent with that of a functional connectivity study [60] suggesting that patients with early AD manifested decreased hippocampal-SFG connectivity compared with healthy controls. The hippocampal–prefrontal cortex circuit plays critical roles in cognitive and emotional regulation and memory consolidation [50]. Another recent study [61] showed decreased hippocampal-SFG connectivity in patients with MCI. The SFG occupies the medial part of the PFC, which plays a critical role in multi-tasking, social cognition, attention, and emotion [62]. A 7T fMRI study [63] revealed decreased hippocampal-SFG connectivity in AD, suggesting that lower MMSE scores were associated with reduced connectivity between the hippocampus and the SFG. Thus, decreased hippocampal-SFG WM connectivity is a potentially important biomarker for the early clinical diagnosis of AD.

### 4.4. Structural Connectivity after Donepezil Treatment in Early AD

To the best of our knowledge, this is the first study evaluating the hippocampus-related structural connectivity in patients with early AD following donepezil treatment. A functional connectivity study found that patients with early AD showed increased right dorsolateral PFC-left dorsolateral PFC connectivity after eight weeks of donepezil treatment [64]. In our study, the patients with early AD showed increased hippocampal-ITG WM connectivity after 6 months of treatment. The ITG plays an important role in semantic verbal fluency, a cognitive function affected early in the onset of AD [65]. A study [66] investigating the cognitive function of ITG in patients with MCI reported that MMSE scores are significantly positively correlated with hippocampal-ITG connectivity. Patients with early AD with a decline in MMSE scores following nine months of donepezil treatment have been found to show decreased volume in the ITG compared with patients with early AD with increased MMSE scores [67]. Treatment with donepezil has been found to reduce the AChE activity in the AD brain by 29% in the temporal cortex [68]. AChE inhibitors have been found to facilitate reorganization in the neural structures that support cognition [64,69]. Improved K-MMSE scores concomitant with increased hippocampal-ITG WM connectivity are potentially attributed to temporal AChE inhibition.

Donepezil activates central cholinergic transmission and enhances the survival of newborn neurons in the hippocampal dentate gyrus [70]. Dong et al. [71] suggested that donepezil treatment reduces beta-amyloid plaques and increases synaptic density. Beta-amyloid deposition has been linked to AD pathology, and it induces multiple biochemical changes in cells, including an increase in cytosolic calcium, which contributes to the downregulation of the expression of glutamate receptors in the postsynaptic membrane [72]. Patients with early AD have been found to show an increase in the serum concentration of brain-derived neurotrophic factor (BDNF) during donepezil treatment [59]. BDNF belongs to the family of nerve growth factors and plays an important role in neuronal survival and synaptic plasticity in the central nervous system [73]. In a previous study, patients with early-to-middle AD showed increased ADAS-Cog scores after the treatment of acupuncture combined with donepezil, and these scores were positively correlated with the fractional amplitude of low-frequency fluctuation values in the ITG [74]. These findings provide evidence suggesting that hippocampal atrophy and decreased hippocampal-SFG WM connectivity may be closely related to AD pathogenesis and that the increased hippocampal-ITG WM connectivity in patients with early AD after donepezil treatment can be attributed to the treatment.

### 4.5. Limitations and Future Directions

This study has some limitations. First, the number of subjects was not large enough to strengthen the statistical power. Therefore, we used a statistical threshold of a *p*-value of less than 0.05 using Bonferroni correction to compensate for this disadvantage, giving a statistically reliable significance level. Second, the potential factors impacting changes in the neuropsychological scores, such as practice or learning effects, were not considered in the current study. The third limitation of this study involves the lack of an in vivo assessment of the pathophysiological biomarker for AD. Another limitation is the short follow-up duration after the donepezil treatment. Therefore, a placebo-controlled study of a large population of patients with early AD and a long-term follow-up are needed to evaluate the time course of treatment change. Such studies should investigate the changes in structural connectivity between mild and moderate AD and between moderate and severe AD in light of the effects of donepezil treatment. In addition, further studies with a larger number of diffusion-encoding gradient directions on DTI indices are needed to gain more reliable information regarding white matter connectivity.

## 5. Conclusions

This study demonstrates variations in hippocampal white matter connectivity in patients with early-stage AD after 6-month donepezil treatment. The increased K-MMSE scores and hippocampal-ITG white matter connectivity in the treated patients could be attributed to the treatment, suggesting that hippocampal-ITG WM connectivity is a potentially important biomarker for donepezil treatment. These findings can be used to develop theories explaining the etiology of early-stage AD and the mode of treatment using anticholinesterases.

## Figures and Tables

**Figure 1 jcm-12-00967-f001:**
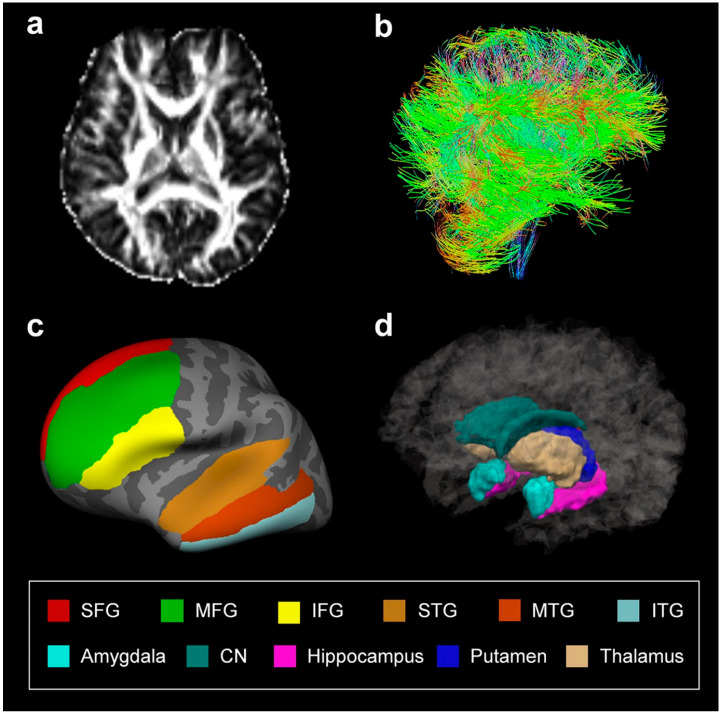
Illustration of diffusion-weighted imaging (**a**), fiber tracts (**b**), cortical regions of interest (ROIs) (**c**), and subcortical ROIs (seed regions) (**d**). SFG; superior frontal gyrus, MFG; middle frontal gyrus, IFG; inferior frontal gyrus, STG; superior temporal gyrus, MTG; middle temporal gyrus, ITG; inferior temporal gyrus, CN; caudate nucleus.

**Figure 2 jcm-12-00967-f002:**
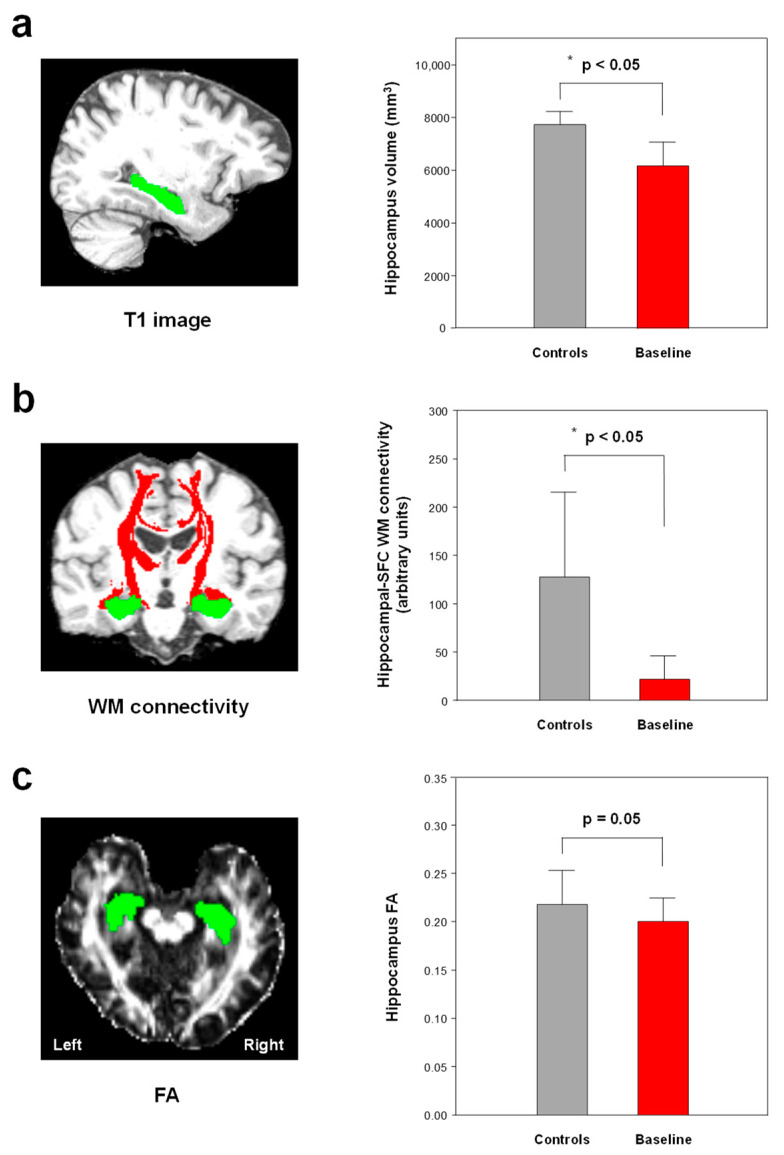
The decreased hippocampal volume (**a**) and hippocampal-superior frontal gyrus (SFG) white matter (WM) connectivity (**b**) in the patients with early AD (baseline) compared to those in the healthy controls (Bonferroni-corrected, *p* < 0.05). The patients with early AD showed decreased fractional anisotropy (FA) in the hippocampus (*p* = 0.05, not corrected for multiple comparisons) (**c**). Green in the left figure, hippocampal seed ROI; red in left figure, WM connectivity. * Meets Bonferroni-corrected significance level (*p* < 0.05).

**Figure 3 jcm-12-00967-f003:**
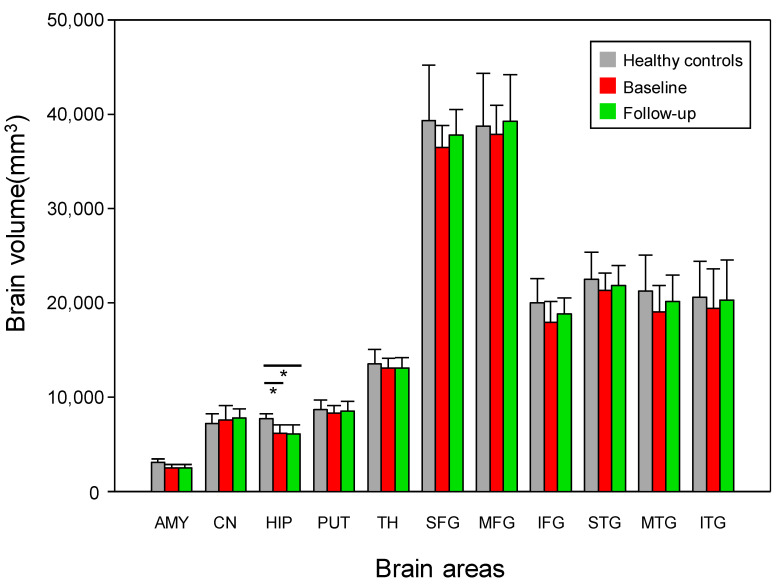
The mean brain volume in the 11 ROIs in the patients with early AD (baseline and 6-month follow-up) and the healthy controls. However, no significant differences were detected in the 11 ROIs before and after donepezil treatment in the patients with early AD. AMY; amygdala, CN; caudate nucleus, HIP; hippocampus, PUT; putamen, SFG; superior frontal gyrus, MFG; middle frontal gyrus, IFG; inferior frontal gyrus, STG; superior temporal gyrus, MTG; middle temporal gyrus, ITG; inferior temporal gyrus. * Meets Bonferroni-corrected significance level (*p* < 0.05).

**Figure 4 jcm-12-00967-f004:**
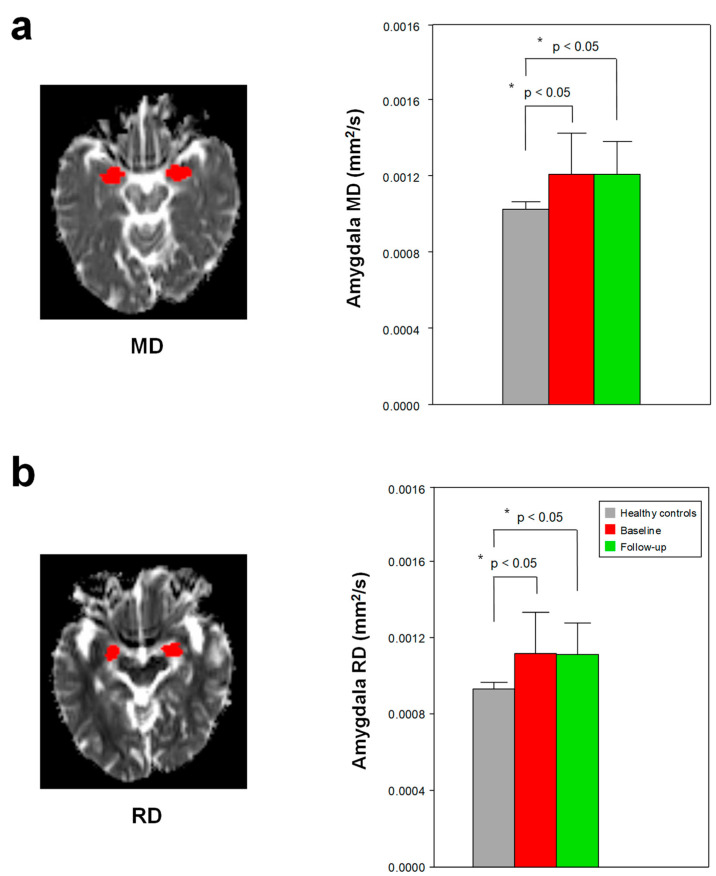
The higher mean diffusivity (MD) (**a**) and radial diffusivity (RD) (**b**) in the amygdala in the patients with early AD compared to those in the healthy controls (Bonferroni-corrected, *p* < 0.05). None of the other 10 ROIs showed significant differences in the DTI scalars between the healthy controls and the patients with early AD. In addition, no significant differences were found between the patients with early AD and the treated patients in the 11 ROIs. Red in the left figure, the amygdala ROI. * Meets Bonferroni-corrected significance level (*p* < 0.05).

**Figure 5 jcm-12-00967-f005:**
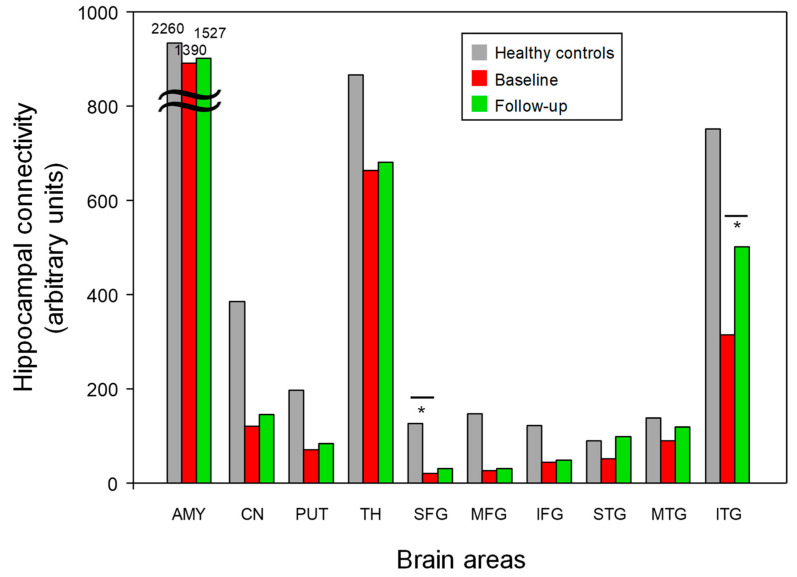
The functional connectivity findings using the hippocampus as a seed region in the patients with early AD (baseline and 6-month follow-up) and the healthy controls. The patients with early AD showed a significant decrease in hippocampal-SFG WM connectivity compared to the healthy controls (Bonferroni-corrected, *p* < 0.05). Following the 6-month donepezil treatment, the patients with early AD showed increased hippocampal-ITG WM connectivity (Bonferroni-corrected, *p* < 0.05). AMY; amygdala, CN; caudate nucleus, HIP; hippocampus, PUT; putamen, SFG; superior frontal gyrus, MFG; middle frontal gyrus, IFG; inferior frontal gyrus, STG; superior temporal gyrus, MTG; middle temporal gyrus, ITG; inferior temporal gyrus. * Meets Bonferroni-corrected significance level (*p* < 0.05).

**Table 1 jcm-12-00967-t001:** Differential brain volumes in the patients with early AD (baseline and 6-month follow-up) and healthy controls.

ROIs	Patients with AD		Healthy Controls (HC)	Statistical Analysis					
	Baseline	6-Month Follow-Up		Baseline vs. Follow-Up		Baseline vs. HC		Follow-Up vs. HC	
				*p*-Value	Cohen’s d	*p*-Value	Cohen’s d	*p*-Value	Cohen’s d
Amygdala	2.5 (0.4)	2.5 (0.4)	3.1 (0.4)	*p* = 0.59	0.31	*p* < 0.05 *	1.70	*p* < 0.05 *	1.70
Caudate nucleus	7.6 (0.4)	7.8 (0.9)	7.2 (1.1)	*p* = 0.31	0.41	*p* = 0.63	0.33	*p* = 0.17	0.68
Hippocampus	6.2 (0.9)	6.1 (0.9)	7.7 (0.5)	*p* = 0.68	0.34	*p* < 0.05 *	2.27	*p* < 0.05 *	2.34
Putamen	8.3 (0.8)	8.5 (1.0)	8.7 (1.0)	*p* = 0.11	0.61	*p* = 0.35	0.45	*p* = 0.45	0.18
Thalamus	13.0 (1.0)	13.1 (1.1)	13.6 (1.5)	*p* = 0.86	0.11	*p* = 0.69	0.43	*p* = 0.45	0.39
SFG	36.4 (2.4)	37.7 (2.7)	39.3 (5.8)	*p* = 0.21	0.72	*p* = 0.20	0.69	*p* = 0.69	0.37
MFG	37.9 (3.0)	39.2 (4.9)	38.7 (5.6)	*p* = 0.52	0.63	*p* = 0.90	0.20	*p* = 0.76	0.11
IFG	17.9 (2.2)	18.8 (1.7)	20.0 (2.6)	*p* = 0.11	0.67	*p* = 0.09	0.91	*p* = 0.31	0.58
STG	21.3 (1.9)	21.8 (2.1)	22.5 (2.9)	*p* = 0.52	0.49	*p* = 0.23	0.53	*p* = 0.51	0.28
MTG	19.0 (2.8)	20.1 (2.8)	21.2 (3.1)	*p* = 0.77	0.51	*p* = 0.15	0.71	*p* = 0.40	0.37
ITG	19.4 (4.2)	20.3 (4.2)	20.6 (3.8)	*p* = 0.21	0.61	*p* = 0.45	0.31	*p* = 0.63	0.07

SFG; superior frontal gyrus, MFG; middle frontal gyrus, IFG; inferior frontal gyrus, STG; superior temporal gyrus, MTG; middle temporal gyrus, ITG; inferior temporal gyrus. * Meets Bonferroni-corrected significance level (*p* < 0.05).Data are presented as mean × 10^3^ (standard deviation × 10^3^)

**Table 2 jcm-12-00967-t002:** The functional connectivity findings using the hippocampus as a seed region in the patients with early AD (baseline and 6-month follow-up) and the healthy controls.

ROIs	Patients with AD		Healthy Controls (HC)	Statistical Analysis					
	Baseline	6-Month Follow-Up		Baseline vs. Follow-Up		Baseline vs. HC		Follow-Up vs. HC	
				*p*-Value	Cohen’s d	*p*-Value	Cohen’s d	*p*-Value	Cohen’s d
Amygdala	13.9 (8.6)	15.3 (14.2)	22.6 (20.2)	*p* = 0.67	0.13	*p* = 0.41	0.62	*p* = 0.33	0.46
Caudate nucleus	1.2 (2.5)	1.5 (1.4)	3.9 (2.5)	*p* = 0.28	0.18	*p* = 0.01	1.16	*p* = 0.03	1.32
Putamen	0.7 (0.6)	0.9 (0.6)	2.0 (1.4)	*p* = 0.33	0.47	*p* = 0.01	1.30	*p* = 0.01	1.16
Thalamus	6.6 (8.2)	6.8 (10.1)	8.7 (7.3)	*p* = 0.86	0.07	*p* = 0.09	0.28	*p* = 0.03	0.23
SFG	0.2 (0.2)	0.3 (0.3)	1.3 (1.2)	*p* = 0.43	0.45	*p* < 0.05 *	1.33	*p* = 0.01	1.20
MFG	0.3 (0.3)	0.3 (0.3)	1.5 (1.6)	*p* = 0.48	0.21	*p* = 0.01	1.19	*p* = 0.02	1.16
IFG	0.5 (0.7)	0.5 (0.4)	1.2 (1.3)	*p* = 0.20	0.16	*p* = 0.06	0.84	*p* = 0.18	0.86
STG	0.5 (0.6)	1.0 (0.8)	0.9 (0.6)	*p* = 0.11	0.77	*p* = 0.08	0.68	*p* = 0.87	0.14
MTG	0.9 (0.8)	1.2 (0.9)	1.4 (1.3)	*p* = 0.51	0.42	*p* = 0.29	0.52	*p* = 0.81	0.19
ITG	3.2 (4.9)	5.0 (4.6)	7.5 (6.9)	*p* < 0.05 *	2.53	*p* = 0.01	0.80	*p* = 0.57	0.47

SFG; superior frontal gyrus, MFG; middle frontal gyrus, IFG; inferior frontal gyrus, STG; superior temporal gyrus, MTG; middle temporal gyrus, ITG; inferior temporal gyrus. * Meets Bonferroni-corrected significance level (*p* < 0.05). Data are presented as mean × 10^2^ (standard deviation × 10^2^).

## Data Availability

The data that support the findings of this study are available from the corresponding author, Gwang-Woo Jeong, upon reasonable request.

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
