# Peer review of "Increased Hippocampal-Inferior Temporal Gyrus White Matter Connectivity following Donepezil Treatment in Patients with Early Alzheimer’s Disease: A Diffusion Tensor Probabilistic Tractography Study"

_jcm, 2023, doi:10.3390/jcm12030967_

Round 1
Reviewer 1 Report
Dear Authors, I read with interest the manuscript entitled “Increased hippocampal-inferior temporal cortex white matter connectivity following donepezil treatment in patients with mild cognitive impairment: A diffusion tensor probabilistic tractography study”. Some points deserve further clarification.
1) The NINCS-ADRDA criteria for AD did not encompass the assessment with in vivo biomarkers to define probable AD. Did patients in this cohort undergo any pathophysiological biomarker for AD? If not, please consider this important limitation and report this lack of in vivo assessment in the limits of the study.
2) Hippocampal atrophy is usually found in AD patients, but also in other neurological diseases, including neurodegenerative disorders, and in some neuropsychiatric conditions. Despite hippocampal damage is typically a key feature of AD, this finding is very unspecific; analogously, the concept of Mild Cognitive Impairment is a broad term, so heterogeous in absence of etiology definition. Accordingly, I recommend to specify whether and what papers focused on MCI - among those reported by authors - considered the etiology of MCI (i.e., if an in vivo biomarkers assessment of underlying pathology was available for MCI subjects). For example, authors report that previous studies showed decreased WM connectivity between hippocampus and SFG in AD patients. I suggest to verify the clinical status of these patients (preclinical, MCI, dementia), since “AD patients” is a too broad definition. Moreover, since etiology of MCI patients in this study could not have been defined by means of in vivo pathophysiological AD biomarkers, it would be important to differentiate the assertion of structural connectivity patterns for MCI in general and for AD patients. I believe this point represents a key feature to be taken into account for the discussion of the main findings of this paper.
3) The CDR total score of 1 is not compatible with MCI definition. Did any patients in the cohort obtain a CDR of 1? If so, these patients may meet criteria for probable AD, but not for MCI. For MCI due to AD, I suggest to refer to NIA-AA criteria of Albert et al, 2011 (Albert MS, DeKosky ST, Dickson D, Dubois B, Feldman HH, Fox NC, Gamst A, Holtzman DM, Jagust WJ, Petersen RC, Snyder PJ, Carrillo MC, Thies B, Phelps CH. The diagnosis of mild cognitive impairment due to Alzheimer's disease: recommendations from the National Institute on Aging-Alzheimer's Association workgroups on diagnostic guidelines for Alzheimer's disease. Alzheimers Dement. 2011 May;7(3):270-9. doi: 10.1016/j.jalz.2011.03.008).
4) Please specify if enrollment criteria for healthy controls excluded other significant neurological or psychiatric conditions.
5) The MMSE and ADAS-Cog are not classifiable as questionnaires, as GDS does, but represent cognitive or neuropsychological scales. The CDR also is a semi-structured interview with patient and informant focusing on cognitive, behavioral and functional domains. Please modify the sentence in Methods section.
6) It might be very interesting to further discuss the hippocampal cortical and subcortical connections within the memory system for learning and recall episodic traces, in particular the relationship with the lateral prefrontal cortex and the inferior temporal lobe. I suggest some reference to be considered.
Eichenbaum H. Prefrontal-hippocampal interactions in episodic memory. Nat Rev Neurosci. 2017 Sep;18(9):547-558. doi: 10.1038/nrn.2017.74.
Li M, Long C, Yang L. Hippocampal-prefrontal circuit and disrupted functional connectivity in psychiatric and neurodegenerative disorders. Biomed Res Int. 2015;2015:810548. doi: 10.1155/2015/810548
Teipel S, Grothe MJ; Alzheimer´s Disease Neuroimaging Initiative. Does posterior cingulate hypometabolism result from disconnection or local pathology across preclinical and clinical stages of Alzheimer's disease? Eur J Nucl Med Mol Imaging. 2016 Mar;43(3):526-36. doi: 10.1007/s00259-015-3222-3.
Schwab S, Afyouni S, Chen Y, Han Z, Guo Q, Dierks T, Wahlund LO, Grieder M. Functional Connectivity Alterations of the Temporal Lobe and Hippocampus in Semantic Dementia and Alzheimer's Disease. J Alzheimers Dis. 2020;76(4):1461-1475. doi: 10.3233/JAD-191113.
7) Please consider these evidence of pharmacological action on hippocampus activation and functioning to be compared with present findings.
Sanchez PE, Zhu L, Verret L, Vossel KA, Orr AG, Cirrito JR, Devidze N, Ho K, Yu GQ, Palop JJ, Mucke L. Levetiracetam suppresses neuronal network dysfunction and reverses synaptic and cognitive deficits in an Alzheimer's disease model. Proc Natl Acad Sci U S A. 2012 Oct 16;109(42):E2895-903. doi: 10.1073/pnas.1121081109.
Zaidel L, Allen G, Cullum CM, Briggs RW, Hynan LS, Weiner MF, McColl R, Gopinath KS, McDonald E, Rubin CD. Donepezil effects on hippocampal and prefrontal functional connectivity in Alzheimer's disease: preliminary report. J Alzheimers Dis. 2012;31 Suppl 3(0 3):S221-6. doi: 10.3233/JAD-2012-120709.
Accordingly, a main result of this study is that increased connectivity in the temporal lobe has been found between the hippocampus and inferior temporal gyrus after donepezil assumption. However, at baseline, the specific finding was a decreased connectivity between the hippocampus and superior frontal gyrus. I suggest to discuss any possible link between these different patterns of neural connectivity.
8) The link between ITG and verbal fluency as reported is too broad. Please specify the type of fluency (i.e.e, semantic fluency), since phonemic fluency is not typically decreased in AD patients and it is sub-served mainly by frontal areas or circuits.
9) Discussion. “The STG occupies the medial part of PFC (mPFC), which plays a critical role in multi-tasking, social cognition, attention, and emotion”. This sentence probably contains a refuse, since maybe authors would refer to Superior Frontal Gyrus (SFG) instead of STG (Superior Temporal Gyrus).
Author Response
1) The NINCS-ADRDA criteria for AD did not encompass the assessment with in vivo biomarkers to define probable AD. Did patients in this cohort undergo any pathophysiological biomarker for AD? If not, please consider this important limitation and report this lack of in vivo assessment in the limits of the study.
♦ We agree and have added the following sentences (see lines 341-345): “The second limitation of this study involves the lack of in vivo assessment of the pathophysiological biomarker for AD. The NINCDS-ADRDA and the DSM-IV-TR criteria for AD are the prevailing diagnostic standards in research; however, they do not encompass the assessment of in vivo biomarkers for defining the probable AD.”
2) Hippocampal atrophy is usually found in AD patients, but also in other neurological diseases, including neurodegenerative disorders, and in some neuropsychiatric conditions. Despite hippocampal damage is typically a key feature of AD, this finding is very unspecific; analogously, the concept of Mild Cognitive Impairment is a broad term, so heterogeous in absence of etiology definition. Accordingly, I recommend to specify whether and what papers focused on MCI - among those reported by authors - considered the etiology of MCI (i.e., if an in vivo biomarkers assessment of underlying pathology was available for MCI subjects). For example, authors report that previous studies showed decreased WM connectivity between hippocampus and SFG in AD patients. I suggest to verify the clinical status of these patients (preclinical, MCI, dementia), since “AD patients” is a too broad definition. Moreover, since etiology of MCI patients in this study could not have been defined by means of in vivo pathophysiological AD biomarkers, it would be important to differentiate the assertion of structural connectivity patterns for MCI in general and for AD patients. I believe this point represents a key feature to be taken into account for the discussion of the main findings of this paper.
♦ We thank the reviewer for the suggestions. Alzheimer’s disease typically progresses in three general stages: early(mild), middle(moderate) and late(severe). To clarify this issue, we have made up for the weak points in the manuscript (see the leading sentence in the Introduction section).
3) The CDR total score of 1 is not compatible with MCI definition. Did any patients in the cohort obtain a CDR of 1? If so, these patients may meet criteria for probable AD, but not for MCI. For MCI due to AD, I suggest to refer to NIA-AA criteria of Albert et al, 2011 (Albert MS, DeKosky ST, Dickson D, Dubois B, Feldman HH, Fox NC, Gamst A, Holtzman DM, Jagust WJ, Petersen RC, Snyder PJ, Carrillo MC, Thies B, Phelps CH. The diagnosis of mild cognitive impairment due to Alzheimer's disease: recommendations from the National Institute on Aging-Alzheimer's Association workgroups on diagnostic guidelines for Alzheimer's disease. Alzheimers Dement. 2011 May;7(3):270-9. doi: 10.1016/j.jalz.2011.03.008).
♦ We appreciate this comment and the following sentence was added (see lines 111-113): “CDR to assess the severity of cognitive impairment (CDR of 0 = clinical normality; 0.5 = very mild dementia; 1 = mild dementia; 2 = moderate dementia; 3 = severe dementia).”
4) Please specify if enrollment criteria for healthy controls excluded other significant neurological or psychiatric conditions.
♦ We reported criteria for healthy controls in the “2.1. Participants” subsection (see lines 105, 106). All healthy controls without a history of neurological or psychiatric disorders participated in the study.
5) The MMSE and ADAS-Cog are not classifiable as questionnaires, as GDS does, but represent cognitive or neuropsychological scales. The CDR also is a semi-structured interview with patient and informant focusing on cognitive, behavioral and functional domains. Please modify the sentence in Methods section.
♦ We appreciate this comment and have reorganized the “2.1. Participants” subsection (see lines 107-113).
6) It might be very interesting to further discuss the hippocampal cortical and subcortical connections within the memory system for learning and recall episodic traces, in particular the relationship with the lateral prefrontal cortex and the inferior temporal lobe. I suggest some reference to be considered.
Eichenbaum H. Prefrontal-hippocampal interactions in episodic memory. Nat Rev Neurosci. 2017 Sep;18(9):547-558. doi: 10.1038/nrn.2017.74.
Li M, Long C, Yang L. Hippocampal-prefrontal circuit and disrupted functional connectivity in psychiatric and neurodegenerative disorders. Biomed Res Int. 2015;2015:810548. doi: 10.1155/2015/810548
Teipel S, Grothe MJ; Alzheimer´s Disease Neuroimaging Initiative. Does posterior cingulate hypometabolism result from disconnection or local pathology across preclinical and clinical stages of Alzheimer's disease? Eur J Nucl Med Mol Imaging. 2016 Mar;43(3):526-36. doi: 10.1007/s00259-015-3222-3.
Schwab S, Afyouni S, Chen Y, Han Z, Guo Q, Dierks T, Wahlund LO, Grieder M. Functional Connectivity Alterations of the Temporal Lobe and Hippocampus in Semantic Dementia and Alzheimer's Disease. J Alzheimers Dis. 2020;76(4):1461-1475. doi: 10.3233/JAD-191113.
♦ Thank you for pointing this out; we reorganized the Discussion section and included the references you suggested.
7) Please consider these evidence of pharmacological action on hippocampus activation and functioning to be compared with present findings.
Sanchez PE, Zhu L, Verret L, Vossel KA, Orr AG, Cirrito JR, Devidze N, Ho K, Yu GQ, Palop JJ, Mucke L. Levetiracetam suppresses neuronal network dysfunction and reverses synaptic and cognitive deficits in an Alzheimer's disease model. Proc Natl Acad Sci U S A. 2012 Oct 16;109(42):E2895-903. doi: 10.1073/pnas.1121081109.
Zaidel L, Allen G, Cullum CM, Briggs RW, Hynan LS, Weiner MF, McColl R, Gopinath KS, McDonald E, Rubin CD. Donepezil effects on hippocampal and prefrontal functional connectivity in Alzheimer's disease: preliminary report. J Alzheimers Dis. 2012;31 Suppl 3(0 3):S221-6. doi: 10.3233/JAD-2012-120709.
Accordingly, a main result of this study is that increased connectivity in the temporal lobe has been found between the hippocampus and inferior temporal gyrus after donepezil assumption. However, at baseline, the specific finding was a decreased connectivity between the hippocampus and superior frontal gyrus. I suggest to discuss any possible link between these different patterns of neural connectivity.
♦ We thank the reviewer for these suggestions. We reorganized the Discussion section.
8) The link between ITG and verbal fluency as reported is too broad. Please specify the type of fluency (i.e.e, semantic fluency), since phonemic fluency is not typically decreased in AD patients and it is sub-served mainly by frontal areas or circuits.
♦ We agree and have added the type of fluency (see page 311).
9) Discussion. “The STG occupies the medial part of PFC (mPFC), which plays a critical role in multi-tasking, social cognition, attention, and emotion”. This sentence probably contains a refuse, since maybe authors would refer to Superior Frontal Gyrus (SFG) instead of STG (Superior Temporal Gyrus).
♦ We have made this change.

Reviewer 2 Report
In general this is an interesting study to investigate the structural connectivity in the hippocampal and other cortical/ subcortical regions before and after donepezil treatment. The main weakness is small sample size but the supplementary file has explained the difficulty of recruitment. Authors have also included the above as the limitations of their study and provided solution to increase statistical discrimination power.
The methodology is detailed described. The tables and graphs are clear. These are the merits of the paper.
I think the results and conclusion are appropriate and justified.
Just a few minor comments as below:
1) Line 189: it states "KMMISE score improved" but no p value given
2) Line196-197 and 225- 226. The meaning of the sentences are not clear and should be rephrased as below: " no significant difference detected before and after treatment in the MCI patient group."
3) Line 281-283; the explanation is not clear to the earlier sentence which states "However, no change in DTI scalars in all brain areas was detected after donepezil treatment". Is this result being expected and why?
Author Response
1) Line 189: it states "KMMISE score improved" but no p value given
♦ We appreciate this comment and the p-value was added (see line 191).
2) Line196-197 and 225- 226. The meaning of the sentences are not clear and should be rephrased as below: " no significant difference detected before and after treatment in the MCI patient group."
♦ We agree and have clarified this issue throughout the manuscript (see lines 197-199 and 226-228).
3) Line 281-283; the explanation is not clear to the earlier sentence which states "However, no change in DTI scalars in all brain areas was detected after donepezil treatment". Is this result being expected and why?
♦ Thank you for your comment. We removed the sentence because the explanation is not clear.
Round 2
Reviewer 1 Report
Dear Authors, I read the updated version of the manuscript entitled “Increased hippocampal-inferior temporal cortex white matter connectivity following donepezil treatment in patients with mild cognitive impairment: A diffusion tensor probabilistic tractography study”. The paper has been ameliorated. However, some points should be further improved.
1) Despite hippocampal damage is typically a key feature of AD, this finding is very unspecific; I recommend to underline this finding.
The term “early AD” might have not a unique meaning. I suggest to further specify if authors do intend the prodromal (early) phase of AD (i.e., Mild Cognitive Impairment). If so, the CDR score of 1 (as included in enrollment criteria) is not compatible with this definition. Could author cite the reference for the “early AD” classification? This represents a crucial point for the better robustness of results.
2) Authors reported the important limitation of lack of in vivo assessment by means of biomarkers for a biologica diagnosis of AD. However, the sentence “The NINCDS‐ ADRDA and the DSM‐IV‐TR criteria for AD are the prevailing diagnostic standards in research” should be changed, since this was true until the biomarkers have been included in the current criteria for AD.
3) Pag. 3. I suggest to report the GDS as independent scale from neuropsychological ones (i.e., questionnaire for depressive symptoms).
4) The structural imaging findings and neuropsychological changes after donepezil treatment are not comparable. It seems that only subtle improvement in MMSE score was observed. So, I suggest to consider other potential factors leading to observed results in neuropsychological scores, i.e. the practice or learning effect. I also suggest to underline the paucity of neuropsychological changes after 6 months.
Author Response
1) Despite hippocampal damage is typically a key feature of AD, this finding is very unspecific; I recommend to underline this finding.
The term “early AD” might have not a unique meaning. I suggest to further specify if authors do intend the prodromal (early) phase of AD (i.e., Mild Cognitive Impairment). If so, the CDR score of 1 (as included in enrollment criteria) is not compatible with this definition. Could author cite the reference for the “early AD” classification? This represents a crucial point for the better robustness of results.
We thank the reviewer for the suggestions. MCI and mild dementia due to AD represent early-stage AD. We appended a couple of sentences along with some references (see lines 39, 40). We searched for the key word, ‘early AD’ in the 640 published papers via PubMed (https://pubmed.ncbi.nlm.nih.gov/).
2) Authors reported the important limitation of lack of in vivo assessment by means of biomarkers for a biologica diagnosis of AD. However, the sentence “The NINCDS‐ ADRDA and the DSM‐IV‐TR criteria for AD are the prevailing diagnostic standards in research” should be changed, since this was true until the biomarkers have been included in the current criteria for AD.
Thank you for your comment. We removed the sentence because the explanation is unclear.
3) Pag. 3. I suggest to report the GDS as independent scale from neuropsychological ones (i.e., questionnaire for depressive symptoms).
We reported the GDS (see lines 116, 117).
4) The structural imaging findings and neuropsychological changes after donepezil treatment are not comparable. It seems that only subtle improvement in MMSE score was observed. So, I suggest to consider other potential factors leading to observed results in neuropsychological scores, i.e. the practice or learning effect. I also suggest to underline the paucity of neuropsychological changes after 6 months.
We appreciate this comment. In the manuscript, there was no significant correlation between changes of MMSE score and hippocampal-ITG WM connectivity before and after donepezil treatment (p=0.529). We added some sentences in the “3.1. Symptom severity”, “3.4. Hippocampal white matter connectivity”, and “4.5. Limitations and future directions” subsections (see lines 195-198, 201-203, and 352-354).
